# Characterization of Magnesium and Zinc Forms of Sodalite Coatings on Ti6Al4V ELI for Potential Application in the Release of Drugs for Osteoporosis

**DOI:** 10.3390/ma16041710

**Published:** 2023-02-17

**Authors:** Mariusz Sandomierski, Wiktoria Stachowicz, Adam Patalas, Karol Grochalski, Wiesław Graboń, Adam Voelkel

**Affiliations:** 1Institute of Chemical Technology and Engineering, Poznan University of Technology, Ul. Berdychowo 4, 60-965 Poznan, Poland; 2Institute of Mechanical Technology, Poznan University of Technology, Ul. Piotrowo 3, 60-965 Poznan, Poland; 3Department of Computer Science, Rzeszow University of Technology, 35-959 Rzeszow, Poland

**Keywords:** titanium alloy, zeolite, drug delivery, osteoporosis

## Abstract

Osteoporosis is the most common metabolic disease of the skeletal system and is characterized by impaired bone strength. This translates into an increased risk of low-energy fractures, which means fractures caused by disproportionate force. This disease is quite insidious, its presence is usually detected only at an advanced stage, where treatment with pharmaceuticals does not produce sufficient results. It is obligatory to replace the weakened bone with an implant. For this reason, it is necessary to look at the possibilities of surface modification used in tissue engineering, which, in combination with the drugs for osteoporosis, i.e., bisphosphonates, may constitute a new and effective method for preventing the deterioration of the osteoporotic state. To achieve this purpose, titanium implants coated with magnesium or zinc zeolite were prepared. Both the sorption and release profiles differed depending on the type of ion in the zeolite structure. The successful release of risedronate from the materials at a low level was proven. It can be concluded that the proposed solution will allow the preparation of endoprostheses for patients with bone diseases such as osteoporosis.

## 1. Introduction

Osteoporosis is the most common metabolic disease of the skeletal system, causing a painless deterioration of its condition [1]. Impaired bone strength results from the progressive destruction of bone mass and is associated with changes in its microstructure [2]. The disease is usually diagnosed at an advanced stage, and its symptoms are osteoporotic fractures, otherwise known as low-energy fractures. The force that caused the fracture is not commensurate with it [3]. Osteoporosis is a global problem, affecting people of all genders, races and ages. The scale of the effects of its occurrence is primarily influenced by the location of the disease and its etiology. Therefore, it is divided into local and generalized (covering the entire skeleton) while in terms of etiology, we distinguish primary (idiopathic and involutional) and secondary forms [1,3].

Despite the large number of factors affecting the possibility of an osteoporotic state, the crucial cause is disturbances in bone metabolism. The disorders are based on the fact that the resorption process—taking place with the participation of osteoclasts—begins to dominate the process of creating a new bone matrix with the help of osteoblasts. As a result, one observes a significant loss in bone mass and an increasingly less effective process of microdamage repair, which in turn is associated with an increased risk of bone fractures [4,5].

A large number of available pharmaceuticals used in therapy cannot be used indefinitely. After the first phase of treatment, correlating with the type of pharmacological agent taken, it is necessary to conduct comprehensive check-ups related to the risk assessment of their impact on the patient’s body. However, the greatest attention should be paid to the fact that osteoporosis is a disease most often diagnosed at an advanced stage, when pharmacological therapy is not effective enough. It is necessary to replace the weakened bone with an implant. Therefore, it is necessary to look at the prospects related to the possibility of modifying the surface of materials intended for their production in order to improve their properties and positively influence the regeneration process and osseointegration [4,6].

The quality of biomaterials used in tissue engineering must be constantly improved due to the tightening requirements that affect the improvement in the functions performed in the patient’s body [7]. The main activities are focused on improving the connections at the interface between the implant and bone tissue, adjusting the shape to individual needs, and above all, the materials covering the surfaces of implants, which have a significant impact on their osseointegration [8].

Many years of observations on titanium and its alloys have shown that they have a long-term significance for medicine [9]. These materials can be implanted into a patient’s body for periods in excess of twenty-five years. They stand out from other biomaterials because they exhibit unique features, i.e., good corrosion resistance and high biotolerance. Titanium and its alloys are characterized by a high tendency of self-passivation, i.e., an increase in the corrosion resistance of the metal as a result of the formation of a thin, tight and well-bound layer of oxides or salts on its surface [8,10,11].

One of the crucial areas of materials engineering is surface engineering. It allows the use of bioceramics, characterized by high biotolerance, corrosion resistance and permanent connection at the interface between the implant and bone tissue. Thanks to these properties, the combination of ceramic and metallic biomaterials allows for high mechanical resistance, and at the same time increased osseointegration. The presence of a protective coating is a kind of barrier to harmful ions, thus limiting the impact of the implant on nearby tissue. With regard to the treatment of osteoporotic conditions, it is interesting that porous zeolite layers can be used to modify titanium materials. Thanks to their application, it will be possible to create drug carriers that can be released from the implant surface [6,12].

Zeolites are hydrated aluminosilicates in crystalline form, which contain mainly calcium or sodium in their structure [13]. They exhibit unique features such as high biocompatibility, large interaction surface and controlled physicochemical properties. The biocompatibility of zeolites has been proven for many zeolite types. Lutzweiler et al. proved that magnesium and calcium zeolite do not have cytotoxic properties [14]. Cytotoxicity studies were also carried out for zinc zeolites. The absence of zinc zeolite toxicity was proven using MCF-7 cells [15]. Additionally, their regular structure, which can be micro-and meso-, as well as macro-porous, and above all their ability to ion exchange, give promising prospects for using them for the controlled delivery of therapeutic agents to target sites in the human body [15,16]. Zeolites containing calcium ions (divalent ions) in their structure are characterized by the ability to interact well with bisphosphonates (BP) [13]. BPs are the most popular anti-resorptive drugs used in the treatment of osteoporosis, showing high bone selectivity compared to other tissues [8]. As a result of ion exchange with the participation of body fluids, these ions (divalent ions) are replaced by sodium and potassium ions. This translates into the possibility of a controlled release of bisphosphonates into a patient’s body as a result of their loss of interaction with the zeolite surface [13]. Materials based on zeolites are classified as bioactive ceramics because they can mimic the mineral component of the bone matrix. After deposition of zeolite layers on implant surfaces, the zeolite film reduces the occurrence of corrosion and significantly improves the osseointegration of the implanted material [17]. So far, it has been proven that materials modified with zeolite layers are characterized by increased alkaline phosphatase activity, a better adhesion of cells to the implants and expression of bone-related genes. In vitro and in vivo studies have shown accelerated growth of hydroxyapatite on the surface of zeolite-modified implants [12,18,19,20,21,22].

In this work, we prepared zeolite (sodalite) layers containing magnesium and zinc ions and investigated their potential as a carrier for the slow release of a drug (risedronate). Both magnesium and zinc are elements used in the preparation of implants [23,24]. Zinc is still perceived as a material that should not be used in biomedical applications. However, recent research has indicated that the release of zinc ions from implants affects healing and stimulates remodeling and new tissue formation [25]. Zinc materials are also beneficial for tissue engineering thanks to their antibacterial properties. The obtained layers have been broadly characterized and the effectiveness of drug sorption on their surface was determined. The final step was to determine how much drug was released at any given time. The release took place in body fluids because the drug was released due to the exchange of magnesium or zinc ions with sodium ions.

## 2. Materials and Methods

### 2.1. Reagents

Titanium alloy (Ti6Al4V) (Φ 8 mm, 4 mm thick), sodium hydroxide, sodium aluminate, sodium silicate, magnesium chloride, zinc chloride, tris(hydroxymethyl)aminomethane (TRIS) (99.8%) and sodium risedronate (RSD) were obtained from Sigma-Aldrich. Hydrochloric acid (36–38%) was obtained from Avantor.

### 2.2. Sodium Zeolite (Sodalite) Coating on Titanium Alloy

The layer was prepared based on a previously described methodology [6]. Initially, the titanium alloy, Ti6Al4V, was subjected to abrasion with 120 grit sandpaper. In the next step, the samples were purified with demineralized water, ethanol and acetone. Then, the samples were placed in a 30% hydrogen peroxide solution at room temperature (3 h) and dried. The in situ hydrothermal crystallization method was used to produce zeolite coatings on the surface of the titanium alloy. Sodium aluminate (1.13 g) was dissolved in the prepared sodium hydroxide solution (20.80 g in 96 g H_2_O). In the next step, the aluminum source was slowly added to 3.32 g of sodium silicate. The mixture was then stirred at room temperature for 30 min. The deposition of the zeolite coating on the Ti6Al4V titanium surface took place at a temperature of 80 °C for 5 h. After this time, the plates were washed with demineralized water and dried at 100 °C for 24 h.

The resulting material was named Ti-Zeo-Na.

### 2.3. Preparation of Magnesium or Zinc Ions Containing Sodalite Coating on Titanium Alloy

The Ti-Zeo-Na material was placed in appropriate 0.5 M metal (magnesium or zinc) chloride solutions (50 mL) for 24 h. During the process, sodium ions were exchanged for magnesium or zinc ions. The process was repeated three times for each type of ion. The material was then washed three times with demineralized water (to remove excess salt) and then dried at 100 °C for 24 h.

The resulting materials were named:Ti-Zeo-Mg;Ti-Zeo-Zn.

### 2.4. Characterization of Materials

#### 2.4.1. Scanning Electron Microscopy (SEM) and Energy Dispersion Spectroscopy (EDS)

SEM analysis is one of the most commonly used methods to determine surface topography. Using this technique, the effectiveness of surface modification of titanium alloys can be confirmed. The effectiveness of the modification in this work will be confirmed by the formation of crystallites, which are visible in the SEM images. EDS analysis allows to determine the amount and distribution of elements on the surface of materials. EDS analysis is especially useful when a new element appears after modification. Mapping a new element confirms the distribution of, for example, a produced layer or a sorbed drug. In this work, the determination of the distribution of the drug is possible on the basis of the distribution of phosphorus (P) ions, which are present in the drug but not in the zeolite layer. Scanning electron microscopy images were obtained with the use of VEGA apparatus (TESCAN, Brno, Czech Republic), also equipped with an EDS analyzer (Bruker, Ettlingen, Germany). The camera creates an image of the analyzed sample by using a focused electron beam, with which the surface is scanned. The electrons interacting with the sample atoms create signals containing information about the surface topography and its composition.

#### 2.4.2. Fourier Transform Infrared Spectroscopy (FT-IR)

FT-IR analysis is one of the best methods for characterizing layers on the surface of titanium alloys. The analysis allows to determine the presence of given functional groups on the surface of the material. Pure titanium alloy does not have any characteristic bands, due to which it is easy to determine the effectiveness of modification and then, for example, drug attachment. FT-IR analysis was performed using a Vertex70 spectrometer (Bruker Optics, Ettlingen, Germany). To perform the analysis, the ATR reflection mode was used with the aid of an attachment of a diamond crystal. The sample was pressed to the crystal. The spectral range was 4000–600 cm^−1^ and the resolution was 4 cm^−1^.

#### 2.4.3. UV-Vis Spectroscopy

UV-Vis spectroscopy is a technique widely used in the analysis of pharmaceuticals. The concentration of a given compound can be easily and quickly determined using this technique. Due to the fact that the drug used in these studies, risedronate, has a characteristic spectrum, both the sorption and release can be clearly determined using this technique. The UV-Vis UV-2600 spectrophotometer (Shimadzu, Kyoto, Japan) was used to determine changes in the concentration of the drug (risedronate) during the sorption process on the modified surfaces of the titanium alloy and its release under the influence of SBF. Measurements were carried out at wavelengths in the range of 240–305 nm with a maximum at 262 nm. During the sorption process, a TRIS-HCl solution was used as the background, while SBF was used during the drug release process. The amount of drug retained was calculated from the curve of risedronate in 0.1 M TRIS-HCl, pH 7.4. The amount of drug released was calculated using the curve of risedronate in SBF.

#### 2.4.4. Optical Surface Profilometry

Surface analysis techniques such as 3D optical profilometry allow a noninvasive and nondestructive approach with very good resolution on the nano/microscale. Here, we demonstrate how coupling noninvasive morphological surface analysis techniques can help to establish the relationship between the modifications and drug sorption and release.

Optical surface profilometry was carried out using Bruker Alicona RL apparatus, which is an optical 3D measurement system containing a microscope and profiler. The structure of surfaces was measured by using a focus variation method. The focus variation method combines the small depth of focus of an optical system with vertical scanning to provide topographical and color information from the variation in focus [26,27]. To perform a complete detection of the surface with a full depth of field, the optic precision is moved vertically along the optical axis while continuously capturing data from the surface. In order to measure the coating thickness and surface topography parameters, first, a small region of the coating was softly removed by using a micro grinding tool (0.3 mm diameter) and then an area of 1.5 × 1.5 mm of the sample was measured. The software used the 3D image, where the region without the coating was a reference, for the coating thickness measurement. For each sample, about 30 profile measurements were made, which were averaged.

The surface topography measurements were performed according to EN ISO standard 25178. The unevenness can be evaluated using the height parameters Pa, Pq, Sa and Sq. Each parameter is classified according to a primary profile (Pa and Pq) and a surface roughness profile (Sa and Sq) in order to evaluate different aspects of the profile. Roughness (Sa and Sq) measurements were taken with a cut-off wavelength of 250 µm. The accuracy of the roughness measurements of the microscope in terms of uncertainty was U = 50 nm.

#### 2.4.5. Nanoindentation

Modern nanomechanical testing instruments enable quantification of mechanical properties from the single crystal/particle level to the finished coating. Testing of drug-releasing coatings using nanomechanical techniques holds potential to develop fundamental knowledge on the structure–property relationships of these coatings.

Nanoindentation measurements were made with a Picodentor HM500 (Fisher, Sindelfingen, Germany) device according to the ISO 14577-1 standard. In the present work, a Vickers tip was used. Loading was increased along the “loading” line until the indentation load became maximal, equal to a force of 300 mN. Ten nanoindentation tests were performed for each sample.

In general, the hardness-equivalent flow stress ratio is well known to be defined by a Tabor coefficient of 3. Therefore, taking into account the classical von Mises plasticity equation, the dependence of “indentation load–indentation depth” may be used to estimating to the coating strength [28,29,30].

### 2.5. Risedronate Loading and Release Experiments

#### 2.5.1. Drug Sorption

In order to test the sorption of the drug, each modified titanium sample was placed in an Eppendorf tube and flooded with the previously prepared drug solution (0.4 mg of risedronate was dissolved in 2 mL of 0.1 M TRIS-HCl solution at pH 7.4). The tubes were then placed on an orbital shaker (200 rpm speed) for 7 days. The changing concentrations of risedronate in the solutions were tested after 3 h and then after 1, 4 and 7 days. For this purpose, the method of UV-Vis spectroscopy was used.

The resulting materials were named:Ti-Zeo-Mg-RSD;Ti-Zeo-Zn-RSD.

Three repetitions were performed for each material.

The sorption process was also carried out for samples with sodium zeolite (Ti-Zeo-Na). However, the attachment of the drug to this surface was ineffective. This confirmed that the drug is retained only on layers containing divalent ions due to the strong interactions with phosphonate groups present in the structure of bisphosphonates [13].

#### 2.5.2. Drug Release

After risedronate sorption occurred, the modified samples were flooded with 1 mL of simulated body fluid (SBF). The drug release process was carried out at a controlled temperature of 36.6 °C. The drug release was tested using UV-Vis spectroscopy at 24 h intervals for a period of 13 days, and then the intervals were extended to 3 days, and then to 7 days due to the insufficient value of the released drug, which was not determinable.

Three repetitions were performed for each material.

The scheme of material preparation and drug release from the zeolite modified titanium surface is shown in Figure 1.

## 3. Results and Discussion

All materials were characterized by SEM analysis (Figure 2). By analyzing the SEM images, it was observed that a porous surface was obtained on all samples, which confirms the correct modification of the titanium alloy with a zeolite coating. It should also be noted that no significant surface change was visible between the images before and after the sorption of risedronate. The lack of changes may indicate that the drug was most likely attached via ions and not precipitated on the modified titanium surface. This is important because the divalent cation–drug interaction is likely to result in controlled release.

Obtaining a surface with the right pore size brings many benefits. A significant part of the bone matrix is deposited directly inside the pores during regenerative processes. Ideal porosity, in addition to providing a physiologically active space for bone growth, also translates into the activation of repair cells, the proper course of oxygen and tissue fluid exchange [31]. It should also be noted that the value of the pore diameter is very important due to the mechanical properties of the obtained material. Too large a pore size and pore differentiation may translate into an increased risk of damage and cracking of the resulting layer [32].

SEM images at higher magnification confirmed the presence of a zeolite structure. For modifications carried out with the use of magnesium zeolites, separate zeolite crystals with a diameter of about 2–3 μm are visible before and after the sorption process. The images for the modification with zinc zeolite look slightly different because the structure creates visibly connected, blurred fragments, without clear zeolite crystals. Anomalies may be due to the precipitation of zinc hydroxide on the surface of the samples as the initial modifications were made with sodium hydroxide.

Mapping with the use of the EDS system enables the analysis of the elements distributed on the surface of the tested sample (Figure 3). Based on the obtained images, we can observe that the individual ions—magnesium and zinc—are distributed over the entire zeolite surface. The images show empty spaces, but when we compare the distribution of magnesium and zinc with the distribution of silicon and aluminum, we can see that they occur in the same places. This may indicate an effective ion exchange in the structure of the aluminosilicates, and not precipitation on the modified surface. The even distribution of divalent ions is very important because through these ions the drug is attached to the surface of the modified alloy and they determine whether the drug will be released in similar doses from the entire surface of the alloy.

The EDS study confirmed the presence of phosphorus (P), derived from the structure of the bisphosphonate, resulting from the drug sorption process on the modified sample surfaces. Risedronate is a bisphosphonate that has a carbon bridge between phosphorus ions. The indicated ions are quite evenly distributed over the entire surface of the sample, which confirms the homogeneity of the material obtained (Figure 3). The distribution of the drug plays an important role because in the development of drug carriers, we strive to ensure that the release takes place to a similar extent from their entire surface, and thus there is no local overdose.

The analyzed amount of zinc significantly exceeds the amount of magnesium (Table 1). This is because the pores on the samples modified with zinc zeolite have been clogged, which can be seen in the SEM photos. Such a large amount of zinc proves that it is not only present in the material due to ion exchange, but also in other forms. Therefore, the drug risedronate is attached mainly to the zeolite surface. In the case of modification with magnesium zeolites, the drug is also present in a significant amount in the free pores. All this translates into a detectable amount of phosphorus ions by the EDS system, which analyzes the sample at a depth of several μm.

The analysis with the use of Fourier transform infrared spectroscopy allowed to confirm the presence of appropriate functional groups derived from the zeolite skeleton and to prove that the drug (risedronate) sorption process was effective.

The presence of the magnesium zeolite layer was confirmed by several characteristic bands (Figure 4). The first two, located at 3300 cm^−1^ and 1640 cm^−1^, are bands derived from stretching and bending vibrations of adsorbed water and hydroxyl groups on the zeolite surface. The key bands are in the range of 1100–600 cm^−1^, as they are attributed to the vibration of the aluminosilicate framework. A wide and strong band in the range of 1100–800 cm^−1^ is assigned to the internal, asymmetric stretching vibrations of Si-O and Al-O in the SiO_4_/AlO_4_ tetrahedron. The bands in the range of 800–650 cm^−1^ come from symmetrical Al-O stretching vibrations in the Si-O-Al. In the spectrum of titanium alloy with magnesium zeolite, changes after the drug sorption process are hardly visible (Figure 5). This may indicate that only a small amount of risedronate is attached to this surface. The spectrum for the modification with the use of zinc zeolite is different, and the differences before and after sorption are very clear. The noticeable difference in the appearance and intensity of the bands on the spectrum may result from the presence of additional compounds. The characteristic bands after sorption on zinc zeolite in the range 1700–1200 cm^−1^ come from risedronate attached to the modified surface of the titanium alloy. The band at 1100 cm^−1^ indicates the presence of the PO_3_ group, attached to a carbon bridge in the bisphosphonate structure [33].

The process of drug sorption onto the modified surfaces of the titanium alloy was carried out for 7 days. Measurements were carried out at the indicated time intervals:3 h;1 day;4 days;7 days.

The measurements were carried out using a UV-Vis spectrophotometer to observe changes in the concentration of the risedronate solution in which the samples were placed. The amount of drug retained on the modified surface of a titanium alloy with the corresponding magnesium/zinc zeolite is shown in Figure 6.

Based on the graph, we can see that the sorption for both modifications is at a different level, although they were subjected to the same modification processes and flooded with the same amount of solution with the drug.

After 3 h, a negligible amount of the drug adhered to the samples with magnesium zeolite, while more than 0.07 mg of the drug was retained on the surface modified with zinc zeolite. The increase in the amount of bisphosphonate attached to the zinc samples was initially abrupt, and then the changes were not so marked. The cause may be clogged pores, meaning the drug was attached in a large amount on the surface, which confirms the assumption based on the FT-IR spectra. The amount of drug retained in the samples with magnesium zeolite was significantly lower than that of zinc and fluctuated at a similar level with the passage of time, increasing only slightly. The lower amount of drug retained is most likely due to the lower amount of magnesium ions incorporated into the zeolite structure, as this translates into a reduction in the efficiency of the drug sorption process. These results confirm the assumption based on the FT-IR spectrum for the modification with magnesium zeolite.

From the above release profiles, it can be seen that the desorption processes for both materials are different (Figure 7). The doses released from the surface of the alloy modified with zinc zeolite are much higher. This may be because during the sorption process the drug was attached primarily to the surface due to the presence of impurities and clogging of the pores, as well as because more drug was retained on this material. The doses released from the surface of the alloy modified with magnesium zeolite are smaller, and this is due to the fact that it has retained less of the drug. However, when we look at the % release relative to the amount of drug retained, the magnesium material releases more drug. This is due to the fact that zinc ions better complex the drug and retain it longer in the zeolite layer [34].

The graphs show the amount of drug released from the modified surfaces of the titanium alloy within 32 days. It should be noted, however, that despite the stabilization of the desorption level, risedronate will continue to be released as a result of ion exchange. Based on the amount adsorbed and released over this period, it was calculated that only:47.04% of the drug was released from the surface modified with magnesium zeolite;24.58% of the drug was released from the surface was modified with zinc zeolite.

A small dose will allow the desorption time to be extended, and thus it will not release too much and cause toxic effects and inflammation through local overdosing. Relating the conducted research to those presented in the literature, the slow release of the drug is an advantage because it occurs through ion exchange, so that the drug will not reattach to the surface. Materials based on calcium phosphate which are used by other research teams are of great interest in tissue engineering, but due to their structure being similar to bone tissue, the drug is resorbed on their surfaces. Therefore, the released amount of drug does not exceed 25% [35,36].

The obtained results indicate that Ti-Zeo-Mg has released 0.05 mg of risedronate while Ti-Zeo-Zn has released 0.09 mg over one month. This amount was released from an area of 0.5 cm^2^. The surface of standard implants is several hundred times larger; hence, more drugs could be released from their surface. In the case of oral therapy (one tablet contains 150 mg of risedronate), about 0.7 mg of the drug reaches the area affected by osteoporosis from one tablet, which is administered once a month (assuming that only half of the bioavailable drug is retained in the skeletal system and the bioavailability of the entire dose is approximately 1%). To match this value, it would be enough to cover only about 7 cm^2^ of the implant with a layer of magnesium zeolite and only about 4 cm^2^ with zinc zeolite. Of course, the bioavailability of the released drug is also important, so this research needs to be continued.

Surface topography has a vital role in bone healing and enhancing biomechanical properties by increasing mechanical retention (interdigitation) and providing a good stress distribution. In cases of poor bone quality and reduced bone volumes, surface roughness is often used in clinical situations to help accelerate and enhance osseointegration and bone interlocking. Otherwise, the implant fixation would be weakened. Increasing surface roughness can, however, via an increased surface area, increase the potential of microbial colonization and provide shelter to bacteria, hence avoiding removal by antibiotics. Research shows that surface topography can indeed influence the rate at which bone is formed next to the surface [37,38,39,40]. Therefore, it is an important parameter that should be measured for this type of coating. Figure 8 shows a microscopic image of the surface topography. A color scale was used to show changes in the height of the surface profile, with the lowest regions being purple and the highest regions being red.

During the nanoindentation tests, the dependence of “indentation load–indentation depth” was determined. Figure 8 shows the averaged curves along with their standard deviation for each type of zeolite layer.

Based on the topography shown in Figure 8, the layer thickness and surface topography parameters were determined. The averaged curves of indentation load–indentation depth from Figure 9 allowed to determine the estimated value of coating strength. The coating thickness, surface topography parameters and estimated coating strength for each the type of layer are shown in Table 2.

Comparing the values of the coating thickness and the indentation depth for Zeo-Mg and Zeo-Mg-RSD samples in Table 2, it can be seen that they are similar. This proves that the magnesium zeolite layers have been broken and the indenter has reached the titanium core material. The analyzed surface topography parameters significantly decrease in value for the comparison of the samples before and after the sorption process of risedronate (Table 2). This is because the pores in the samples modified zeolite have been clogged. The studies have shown that the optimal Sa surface topography parameter needs to be around 1–1.5 µm because of enhanced osseointegration and bone interlocking of the implant [41]. The obtained significant roughness parameters of the zeolite coating were higher and amounted to 4.42 μm for Zeo-Mg, 2.85 μm for Zeo-Mg-RSD, 3.90 μm for Zeo-Zn and 2.98 μm for Zeo-Zn-RSD. Nevertheless, another study presented a finding that an increase in surface roughness inhibited the adhesion of bacteria [41]. Therefore, the affinity for cell surface adhesion should be checked in other cell assays.

## 4. Conclusions

The titanium alloy Ti6Al4V was successfully modified with zeolite layers containing divalent ions, i.e., magnesium and zinc. The performed analyses allowed us to characterize the obtained coatings and confirm the attachment of the drug (risedronate) to the modified surfaces and confirm the theoretical assumptions.

The formation of porous zeolite layers on titanium surfaces was confirmed. Appropriately sized pores will allow for the improvement of the regeneration process of the tissues surrounding the implant with the obtained covering material. The ions on the modified surfaces are quite evenly distributed, which proves the homogeneity of the materials produced. Additionally, it can be seen that the surface topography for the Zeo-Mg and Zeo-Zn coatings are similar and the sorption of the drug on their surface causes the Sa and Sq parameters to drop to similar values.

Although all samples underwent the same modification process and were placed in an identical amount of drug solution, the sorption and release profile for each modification were different. Thanks to the use of various divalent ions, we can influence the drug release profile and control the time taken for, as a result of ion exchange, the drug to be released into the body to the place directly affected by the disease state. As the drug is released slowly and gradually, not too much will be released, which could irritate the surrounding tissues, cause local overdose and cause inflammation.

In samples containing magnesium ions, the drug risedronate attached via ions mainly inside the pores of the zeolite material. On the other hand, on the surfaces with zinc zeolite, due to the clogging of the pores, most likely with zinc ions in the form of hydroxides, the bisphosphonate was attached mainly to the surface of the samples. A different mechanism of drug sorption between Mg and Zn may explain the different changes in the coating strength measured in nanoidentification tests.

In the case of zeolite modification with zinc ions, the methodology should be worked on to reduce the possible amount of impurities present. However, this does not preclude the use of this material. It should be highlighted that this material may be useful due to the faster release of the drug, so cell studies should be carried out to confirm this assumption. It may also have antibacterial properties. In addition, the higher estimated strength of the Zeo-Zn-RSD coating may suggest that it will not be damaged during the mechanical fixation of the endoprosthesis in the bone. The use of zinc ions may seem undesirable due to the many studies describing its toxicity. However, the use of these ions in this type of material can bring many benefits. The first important property is the ability to obtain materials with antibacterial properties [42]. In addition, these ions have a confirmed effect in the case of supporting the treatment of osteoporosis, so, together with a bisphosphonate, they can cause an improved double therapeutic effect [43].

These studies should be continued for other ions as well. Another cation that can also be used in drug carriers (especially those that will be used in implants) is strontium. This is because strontium has the ability to inhibit bone resorption and promote bone formation [44,45].

The successive release of risedronate over many months will allow for the preparation of endoprostheses, which, apart from replacing the weakened bone, may inhibit the further progression of the osteoporotic state in their environment.

## Figures and Tables

**Figure 1 materials-16-01710-f001:**
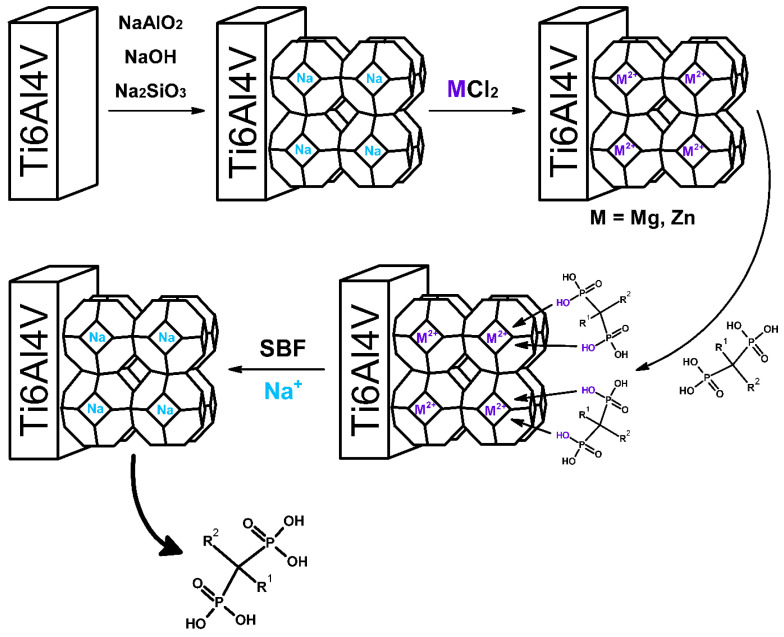
Scheme of drug release from titanium alloy modified with magnesium and zinc zeolite obtained in this work.

**Figure 2 materials-16-01710-f002:**
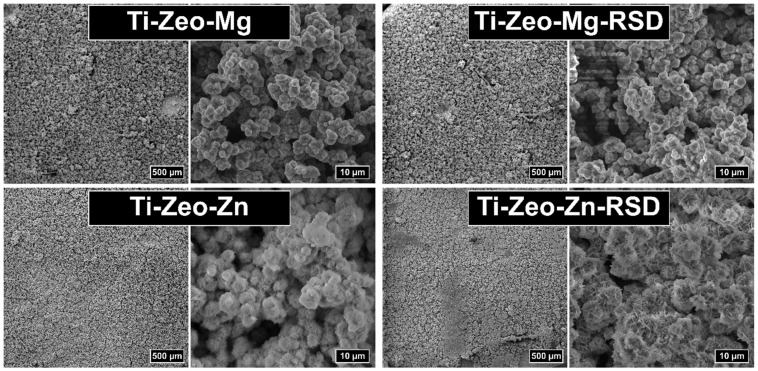
SEM images of the alloy with magnesium and zinc zeolite layers before and after drug adsorption.

**Figure 3 materials-16-01710-f003:**
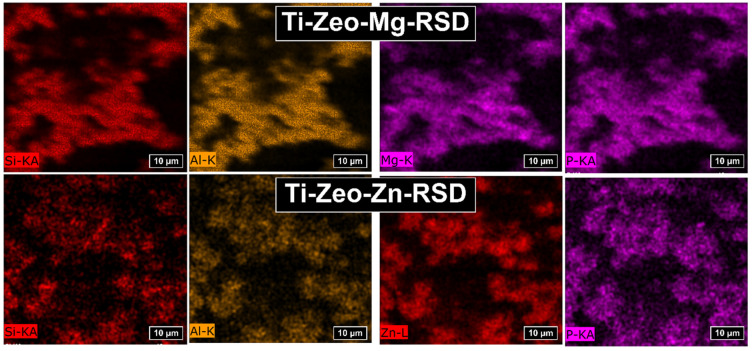
Distribution of ions in the synthesized layers.

**Figure 4 materials-16-01710-f004:**
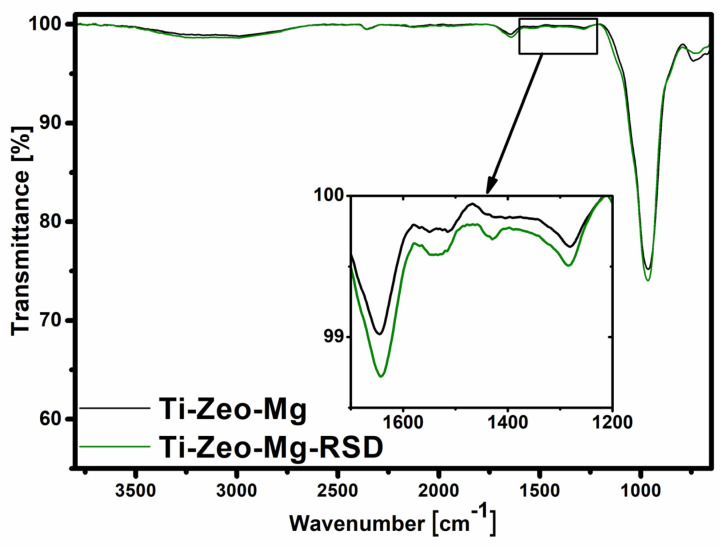
FTIR spectra of magnesium zeolite layers before and after drug sorption.

**Figure 5 materials-16-01710-f005:**
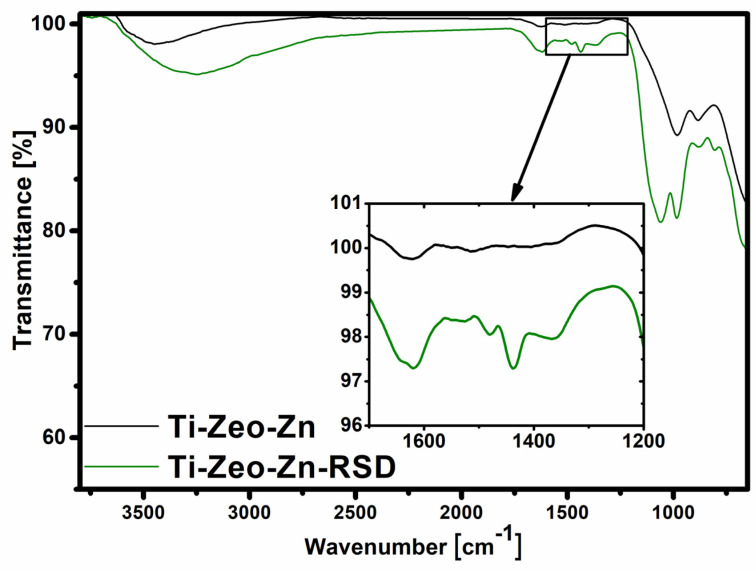
FTIR spectra of zinc zeolite layers before and after drug sorption.

**Figure 6 materials-16-01710-f006:**
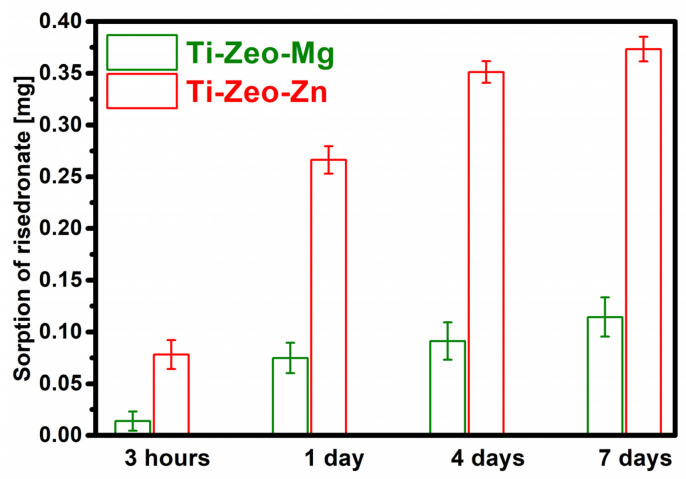
Amount of drug retained on prepared materials (the graph shows the mean values and standard deviation) (three repetitions were performed for each material).

**Figure 7 materials-16-01710-f007:**
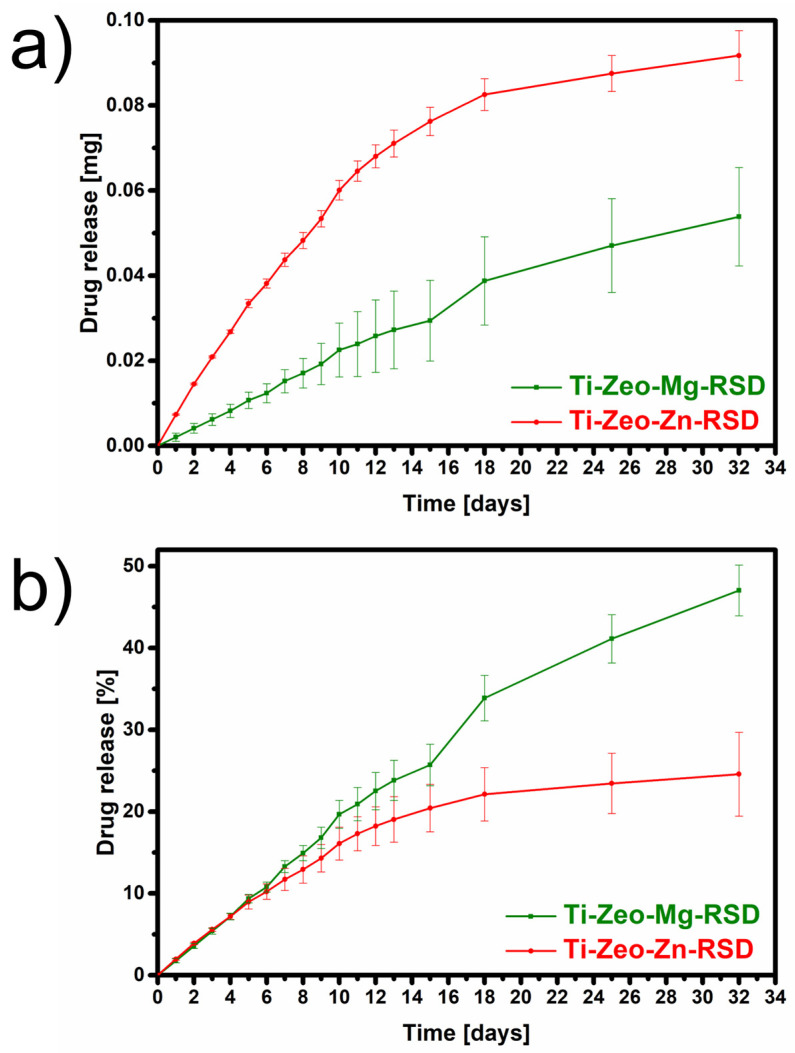
The amount of the drug released from the prepared materials. The cumulative amount of drug released from the material in milligrams (**a**) and in % (**b**) (three repetitions were performed for each material).

**Figure 8 materials-16-01710-f008:**
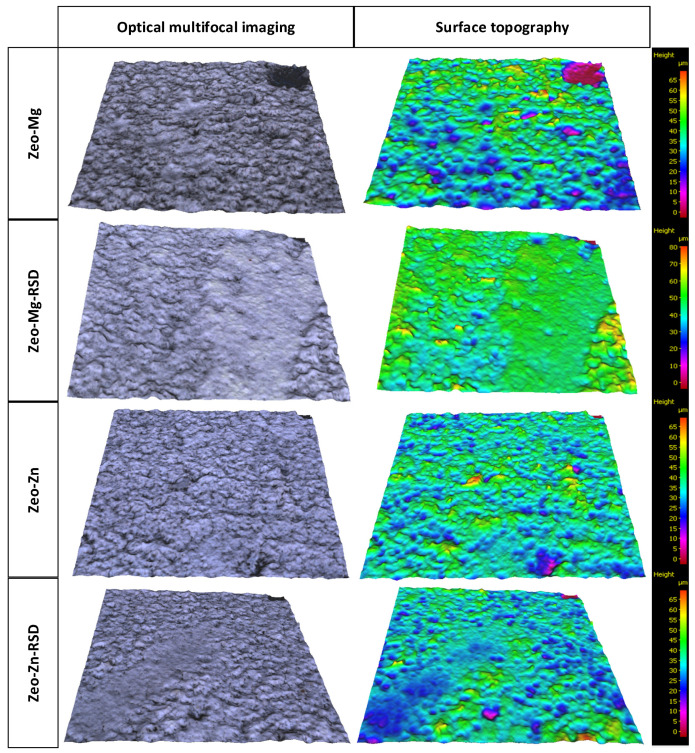
The multifocal images with surface topography of magnesium and zinc zeolite layers before and after drug adsorption (three samples were measured for each material).

**Figure 9 materials-16-01710-f009:**
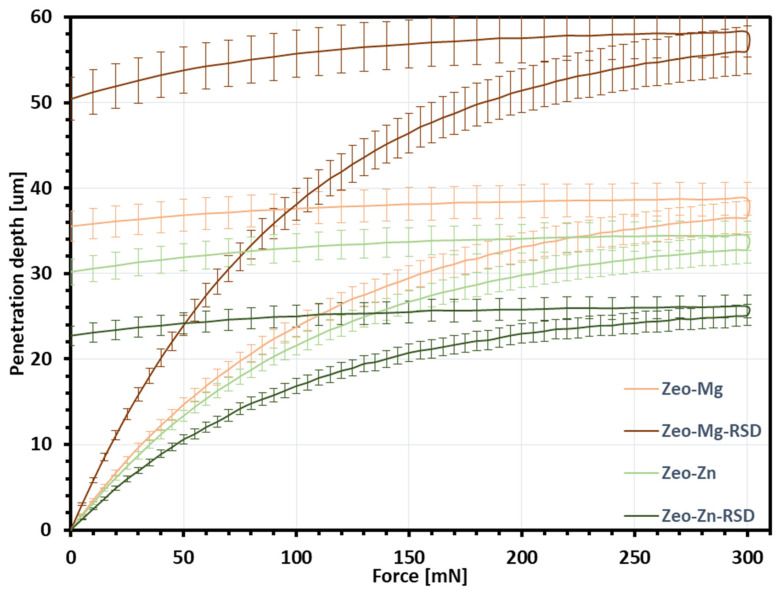
The averaged curves of indentation load–indentation depth for each type of layer (for each material, three samples were measured with five repetitions of the nanoindentation test).

**Table 1 materials-16-01710-t001:** The content of elements in materials after drug sorption (wt. %).

	Ti-Zeo-Mg-RSD	Ti-Zeo-Zn-RSD
Mg	0.66 ± 0.24	-
Zn	-	31.67 ± 5.99
P	1.81 ± 1.10	6.14 ± 2.04
N	-	1.75 ± 1.60
Al	11.00 ± 5.88	6.16 ± 1.69
Si	12.44 ± 6.62	3.04 ± 1.90

**Table 2 materials-16-01710-t002:** Coating thickness, surface topography parameters and estimated coating strength for each type of layers.

	Coating Thickness (μm)	Profile and Surface Topography Parameters				Indentation Depth (μm)	Estimated Coating Strength (MPa)
		Pa (μm)	Pq (μm)	Sa (μm)	Sq (μm)		
Ti-Zeo-Mg	37.7	6.46	8.26	4.42	5.80	37.0	9.04
Ti-Zeo-Mg-RSD	58.5	4.95	5.05	2.85	3.99	55.9	3.95
Ti-Zeo-Zn	40.4	5.25	6.80	3.90	5.08	33.3	12.09
Ti-Zeo-Zn-RSD	34.9	4.51	5.76	2.98	3.96	25.0	21.25

Pa—arithmetic mean primary profile; Pq—squared mean primary profile; Sa—arithmetical mean surface roughness; Sq—squared mean surface roughness.

## Data Availability

Not applicable.

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
