# Peer review of "Characterization of Magnesium and Zinc Forms of Sodalite Coatings on Ti6Al4V ELI for Potential Application in the Release of Drugs for Osteoporosis"

_materials, 2023, doi:10.3390/ma16041710_

Round 1
Reviewer 1 Report
In this study, the authors prepared zeolite layers containing magnesium and zinc ions. The layers have been broadly characterized and the effectiveness of drug sorption on their surface was determined. This study concluded that the potential application of sodalite coatings on Ti6Al4V ELI in the release of drug for osteoporosis.
In general, the authors validate their hypothesis throughout the experiments, however, some concerns should be fully address prior to acceptance.
1. The inconsistency of decimal point in Table 1.
2. Please included the N number in each experiment, e.g., Figure 7. Figure 8.
3. Is the drug absorbed on the zeolite layers comparable to clinical need?
4. Is the amount of drug released from the zeolite layers enough as a medication for osteoporosis management?
5. Please keep the consistency of reference format (e.g., the capital letter of article name).
Author Response
Dear Reviewer, thank you for all your comments. Undoubtedly, your comments allowed us to improve the quality of our work. Below we send the answers and we hope that now you will agree to accept our work.
Comment 1: The inconsistency of decimal point in Table 1.
Response 1: Dear reviewer, we apologize for this mistake. This has been corrected.
Comment 2: Please included the N number in each experiment, e.g., Figure 7. Figure 8.
Response 2: Information has been added to each technique.
Comment 3: Is the drug absorbed on the zeolite layers comparable to clinical need?
Response 3: Dear Reviewer, the obtained values indicate that the quantities are sufficient. The following information has been added:
- “The obtained results indicate that Ti-Zeo-Mg has released 0.05 mg of risedronate while Ti-Zeo-Zn has released 0.09 mg during one month. This amount was released from an area of 0.5 cm2. The surface of standard implants is several hundred times larger due to which more drugs could be released from their surface. In the case of oral therapy (tablet contains 150 mg of risedronate), about 0.7 mg of the drug reaches the area affected by osteoporosis from one tablet, which is administered once a month (assuming that only half of the bioavailable drug is retained in the skeletal system and the bioavailability of the entire dose is approximately 1%). To achieve this value, it would be enough to cover only about 7 cm2 of the implant with a layer of magnesium zeolite and only about 4 cm2 with zinc zeolite. Of course, the bioavailability of the re-leased drug is also important, so this research needs to be continued.”
Comment 4: Is the amount of drug released from the zeolite layers enough as a medication for osteoporosis management?
Response 4: The answer is given in the previous question.
Comment 5: Please keep the consistency of reference format (e.g., the capital letter of article name).
Response 5: The formatting of the publication has been corrected using the "Zotero" program, which is recommended by the MDPI.
Reviewer 2 Report
This report is related to drug release near bone damage areas, let's describe the need of using zinc.
The toxicity results of materials need to provide, especially for zinc content.
Zinc materials can be toxic if it shows toxicity.
Provide alternative solutions to avoid toxicity damage to bone and tissues.
Materials and methods lack sufficient details.
Results and discussion need to reflect scientific soundness.
Avoid just describing results, provide scientific descriptions and understandings.
In conclusion, suggest possible alternative materials to replace zinc in the composite.
Author Response
Dear Reviewer, thank you for all your comments. Undoubtedly, your comments allowed us to improve the quality of our work. Below we send the answers and we hope that now you will agree to accept our work.
Comment 1: This report is related to drug release near bone damage areas, let's describe the need of using zinc.
Response 1: Dear Reviewer, we agree that this was not sufficiently emphasized in the work, which is why we have added fragments on this subject in the introduction and summary.
- “Both magnesium and zinc are elements used in the preparation of implants [22,23]. Zinc is still perceived as a material that should not be used in biomedical applications. However, recent research indicates that the release of zinc ions from implants affects healing and stimulates remodeling and new tissue formation [24]. Zinc materials are also beneficial for tissue engineering thanks to their antibacterial properties.”
- “The use of zinc ions may seem undesirable due to the many studies describing its toxicity. However, the use of these ions in this type of material can bring many benefits. The first important property is the ability to obtain materials with antibacterial properties [37]. In addition, these ions have a confirmed effect in the case of supporting the treatment of osteoporosis, so together with a bisphosphonate they can cause a better double therapeutic effect [38].”
Comment 2: The toxicity results of materials need to provide, especially for zinc content.
Response 2: Dear Reviewer, of course such studies are important, but at the moment we base the lack of toxicity on literature reports. Due to this, we have added the following text in our work:
- “The biocompatibility of zeolites has been proven for many types. Lutzweiler et al. proved that magnesium and calcium zeolite do not have cytotoxic properties [14]. Cy-totoxicity studies were also carried out for zinc zeolites. The absence of zinc zeolite toxicity was proven using MCF-7 cells [15].”
In the future, we plan to perform such tests on a large scale for our materials, but the studies presented in this paper are preliminary studies. Following the publication of this paper, we hope to receive funding to continue this research and extend it to cell research.
Comment 3: Zinc materials can be toxic if it shows toxicity.
Response 3: Dear Reviewer, we hope we have clarified everything in the previous answers.
Comment 4: Provide alternative solutions to avoid toxicity damage to bone and tissues.
Response 4: In our opinion, this will not happen. However, the first solution is the use of magnesium ions, which we described in the paper. Another could be the use of, for example, strontium ions and we have added information on this in the summary:
- “These studies should be continued for other ions as well. Cation that can be also used in drug carriers (especially those that will be used in implants) is strontium. This is because strontium has the ability to inhibit bone resorption and promote bone formation [43,44].”
Comment 5: Materials and methods lack sufficient details.
Response 5: The "Materials and Methods" section has been extended.
- “SEM analysis is one of the most commonly used methods to determine the surface topography. Using this technique, the effectiveness of surface modification of titanium alloys can be confirmed. The effectiveness of the modification in this work will be confirmed by the formation of crystallites, which are visible in the SEM images. EDS analysis allows to determine the amount and distribution of elements on the surface of materials. EDS analysis is especially useful when a new element appears after modification. Mapping a new element confirms the distribution of, for example, a produced layer or a sorbed drug. In this work, the determination of the distribution of the drug is possible on the basis of the distribution of phosphorus (P) ions, which are present in the drug, but not in the zeolite layer.“
- “FT-IR analysis is one of the best methods for characterizing layers on the surface of titanium alloys. The analysis allows to determine the presence of given functional groups on the surface of the material. Pure titanium alloy does not have any characteristic bands due to which it is easy to determine the effectiveness of modification and then, for example, drug attachment.”
- “UV-Vis spectroscopy is a technique widely used in the analysis of pharmaceuticals. The concentration of a given compound can be easily and quickly determined using this technique. Due to the fact that the drug used in these studies, risedronate, has a characteristic spectrum, both sorption and release can be clearly determined using this technique.”
- “Surface analysis techniques like 3D optical profilometry allow noninvasive and nondestructive approach with very good resolution at nano-microscale. Here are demonstrated, how coupling noninvasive morphological surface analysis techniques can help to establish the relationship between the modifications on drug surption and release.”
- “Each parameter is classified according to primary profile (Pa, Pq) and surface rough-ness profile (Sa, Sq) in order to evaluate different aspects of the profile.”
- “Modern nanomechanical testing instruments enable quantification of mechanical properties from the single crystal/particle level to the finished coatings. Testing of drug realeasing coatings using nanomechanical techniques holds potential to develop fundamental knowledge in the structure–property relationships of these coatings.”
Comment 6: Results and discussion need to reflect scientific soundness.
Response 6: The results and discussion have been improved. For example:
- “This is important because the divalent cation-drug interaction is likely to result in con-trolled release.”
- “The even distribution of divalent ions is very important because through these ions the drug is attached to the surface of the modified alloy and they determine whether the drug will be released in similar doses from the entire surface of the alloy.”
- “The obtained results indicate that Ti-Zeo-Mg has released 0.05 mg of risedronate while Ti-Zeo-Zn has released 0.09 mg during one month. This amount was released from an area of 0.5 cm2. The surface of standard implants is several hundred times larger due to which more drugs could be released from their surface. In the case of oral therapy (tablet contains 150 mg of risedronate), about 0.7 mg of the drug reaches the area affected by osteoporosis from one tablet, which is administered once a month (assuming that only half of the bioavailable drug is retained in the skeletal system and the bioavailability of the entire dose is approximately 1%). To achieve this value, it would be enough to cover only about 7 cm2 of the implant with a layer of magnesium zeolite and only about 4 cm2 with zinc zeolite. Of course, the bioavailability of the re-leased drug is also important, so this research needs to be continued.”
- From the above release profiles, it can be seen that the desorption process for both materials is different (Figure 8). The doses released from the surface of the alloy modified with zinc zeolite are much higher. This may be since during the sorption process the drug was attached primarily to the surface due to the presence of impurities and clogging of the pores as well as more drug retained on this material. The doses released from the surface of the alloy modified with magnesium zeolite are smaller and this is due to the fact that it has retained less of the drug. However, when we look at the % release relative to the amount of drug retained, the magnesium material releases more drug. This is due to the fact that zinc ions better complex the drug and keep it longer in the zeolite layer [34].
Comment 7: Avoid just describing results, provide scientific descriptions and understandings.
Response 7: Dear Reviewer, we have corrected and improved the results description. For example:
- “This is important because the divalent cation-drug interaction is likely to result in con-trolled release.”
- “The even distribution of divalent ions is very important because through these ions the drug is attached to the surface of the modified alloy and they determine whether the drug will be released in similar doses from the entire surface of the alloy.”
- “The obtained results indicate that Ti-Zeo-Mg has released 0.05 mg of risedronate while Ti-Zeo-Zn has released 0.09 mg during one month. This amount was released from an area of 0.5 cm2. The surface of standard implants is several hundred times larger due to which more drugs could be released from their surface. In the case of oral therapy (tablet contains 150 mg of risedronate), about 0.7 mg of the drug reaches the area affected by osteoporosis from one tablet, which is administered once a month (assuming that only half of the bioavailable drug is retained in the skeletal system and the bioavailability of the entire dose is approximately 1%). To achieve this value, it would be enough to cover only about 7 cm2 of the implant with a layer of magnesium zeolite and only about 4 cm2 with zinc zeolite. Of course, the bioavailability of the re-leased drug is also important, so this research needs to be continued.”
- From the above release profiles, it can be seen that the desorption process for both materials is different (Figure 8). The doses released from the surface of the alloy modified with zinc zeolite are much higher. This may be since during the sorption process the drug was attached primarily to the surface due to the presence of impurities and clogging of the pores as well as more drug retained on this material. The doses released from the surface of the alloy modified with magnesium zeolite are smaller and this is due to the fact that it has retained less of the drug. However, when we look at the % release relative to the amount of drug retained, the magnesium material releases more drug. This is due to the fact that zinc ions better complex the drug and keep it longer in the zeolite layer [34].
Comment 8: In conclusion, suggest possible alternative materials to replace zinc in the composite.
Response 8: Information has been added:
- “These studies should be continued for other ions as well. Cation that can be also used in drug carriers (especially those that will be used in implants) is strontium. This is because strontium has the ability to inhibit bone resorption and promote bone formation [43,44].”